# Highlights of the results from the GRAPES-3 experiment

B. Hariharan[1*], S. Ahmad[2], M. Chakraborty[1], S.R. Dugad[1], U.D. Goswami[3], S.K. Gupta[1],
Y. Hayashi[4], P. Jagadeesan[1], A. Jain[1], P. Jain[5], S. Kawakami[4], H. Kojima[6], S. Mahapatra[7],
P.K. Mohanty[1], R. Moharana[8], Y. Muraki[9], P.K. Nayak[1], T. Nonaka[10], A. Oshima[6],
D. Pattanaik[1], B.P. Pant[8], M. Rameez[1], K. Ramesh[1], L.V. Reddy[1], S. Shibata[6], F. Varsi[5],
M. Zuberi[1]

(GRAPES-3 Collaboration)

**1** Tata Institute of Fundamental Research, Homi Bhabha Road, Mumbai 400005, India
**2** Aligarh Muslim University, Aligarh 202002, India
**3** Dibrugarh University, Dibrugarh 786004, India
**4** Graduate School of Science, Osaka City University, Osaka 558-8585, Japan
**5** Indian Institute of Technology Kanpur, Kanpur 208016, India
**6** College of Engineering, Chubu University, Kasugai, Aichi 487-8501, Japan
**7** Utkal University, Bhubaneshwar 751004, India
**8** Indian Institute of Technology Jodhpur, Jodhpur 342037, India
**9** Institute for Space-Earth Environmental Research, Nagoya University, Nagoya 464-8601, Japan
**10** Institute for Cosmic Ray Research, Tokyo University, Kashiwa, Chiba 277-8582, Japan
*89hariharan@gmail.com

October 10, 2022

## Abstract

The GRAPES-3 experiment is a unique, extensive air shower experiment consisting of 400 scintillator detectors spread over $25000\,\mathrm{m}^2$ and a $560\,\mathrm{m}^2$ muon telescope. The experiment located at Ooty, India, has been collecting data for the past two decades. The unique capabilities of GRAPES-3 have allowed the study of cosmic rays over energies from a few TeV to tens of PeV and beyond. The measurement of the directional flux of muons ($E_\mu \geq 1\,\mathrm{GeV}$) by the large muon telescope permits an excellent gamma-hadron separation, which then becomes a powerful tool in the study of multi-TeV gamma-ray sources and the composition of primary cosmic rays. However, the high precision measurements also enable studies of transient atmospheric and interplanetary phenomena such as those produced by thunderstorms and geomagnetic storms. This paper presents some exciting new and recent results, including updates on various ongoing analyses.

# 1 Introduction

Cosmic rays (CRs) were discovered by V.F. Hess more than a century ago. Historically many experiments have studied them in the extraordinary energy range of 100 MeV–100 EeV to understand the origin and properties of CRs. The broad span of the primary energy spectrum has a power-law dependence with multiple spectral breaks namely the knee and ankle at various energies. CRs are predominantly composed of protons ($\sim$90%), helium ($\sim$9%), and heavier elements up to iron, attributing to the remaining 1%. The energy spectrum and nuclear mass composition studies are the primary objectives of any CR experiment. Cosmic ray detection can be broadly classified into two categories, namely direct and indirect detection methods. CRs can be detected directly using detectors aboard space probes and balloon flights. However, this is only possible up to 100 TeV. Beyond this energy, the direct observation is limited by the rapidly falling flux of CRs, detector size, and exposure time. Above 100 TeV, CRs can be detected indirectly by using the extensive air shower (EAS) phenomenon in which the primary cosmic ray (PCR) develops into a shower of particles in the Earth's atmosphere that can be detected at ground level using an array of particle detectors. The GRAPES-3 experiment consists of an array of plastic scintillators and a large area tracking muon telescope for CR studies in a broad energy range of 1 TeV–10 PeV. Because of its tightly packed configuration with a sensitive area of $\sim$2%, larger than other experiments ($<$1%), the energy threshold is brought down as low as 1 TeV. Also, the muon telescope helps to differentiate gamma and hadron-initiated EASs for composition and gamma-ray astronomy studies. The GRAPES-3 energy spectrum and mass composition measurements may provide good overlap with the measurements from direct and indirect detection methods by other experiments. GRAPES-3 can look into the northern and southern hemispheres with reasonably good coverage. The scientific objectives of GRAPES-3 span into multi-energy domains such as atmospheric acceleration, solar phenomena, energy spectrum and composition studies of CRs, and multi-TeV gamma-ray astronomy. This paper discusses published and preliminary results from the various ongoing analyses.

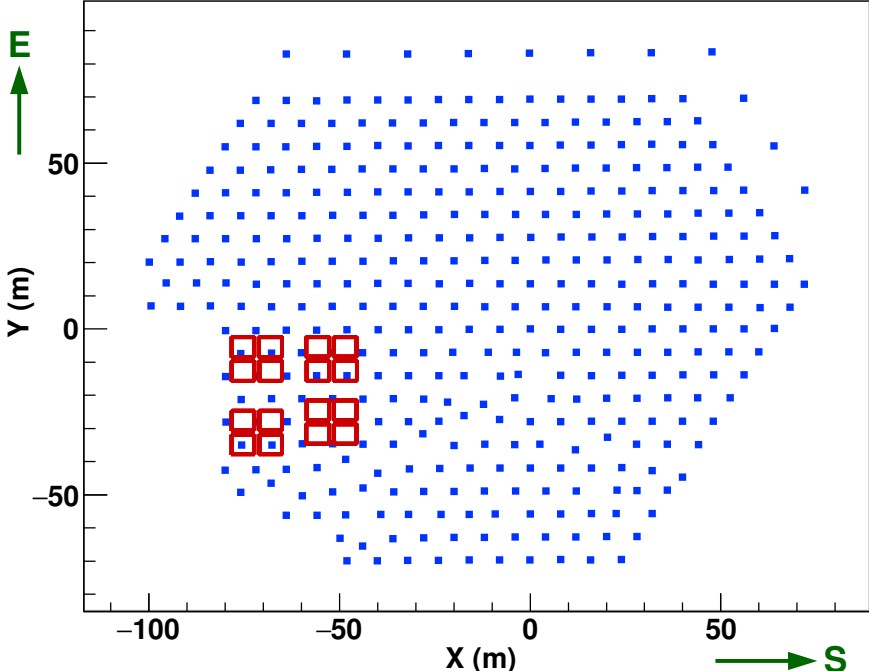

Figure 1: The schematic of the GRAPES-3 experiment consists of an array of plastic scintillators (■) and muon telescope modules (□).

## 2 The GRAPES-3 experiment

The **G**amma **R**ay **A**stronomy at **P**eV Energie**S** – **3** (GRAPES-3) is a ground-based EAS experiment. It is located at Ooty, India (11.4°N, 76.7°E) at an altitude of 2200 m above mean sea level. The near-equatorial placement of the experiment provides a unique advantage for measurements covering both northern and southern hemispheres significantly. GRAPES-3 consists of two detector elements, namely (i) a high-density large area EAS array and (ii) large area tracking muon telescope (G3MT) as shown in Figure 1. The EAS array is designed using 400 plastic scintillators, each with an effective area of 1 m². The scintillators are placed in a hexagonal geometry with an inter-detector separation of 8 m as seen in Figure 1, covering an area of 25000 m². Due to its tightly packed detector configuration, detecting PCRs of energy from 1 TeV to 10 PeV is possible. Each scintillator records the energy deposit and first arrival time of the passing particle with respect to an EAS trigger. This information can be reconstructed offline to get the properties of the PCRs. A detailed report on the detector and data recording system can be found here [1]. Everyday, the GRAPES-3 records about 3.5 million EASs in the above-mentioned energy range.

The second detector element is G3MT which is built using 16 muon telescopic modules as shown in Figure 1. Each muon module has an area of 35 m². Proportional counters (PRCs) are gaseous detectors used to build the G3MT. Each PRC is a 600×10×10 cm³ mild steel tube which is sealed and filled with a P10 gas mixture (90% argon and 10% methane). A 100 $\mu$m thick tungsten wire at the center acts as an anode, whereas the metal body is the cathode. Each muon module consists of four layers of PRCs. Each layer is arranged with 58 PRCs. A 15 cm thick concrete slab sandwiches the layers with the alternate layers placed orthogonal to each other. This configuration allows reconstruction of the detected muons in 169 directions that can be used for physics studies. Above the topmost PRC layer, 2 m thick concrete slabs are placed in an inverted pyramidal shape to provide an energy threshold of $\sec(\theta)$ GeV for muons coming at zenith $\theta$. The primary role of G3MT is to measure the muon content from

the EAS, which is an excellent proxy for differentiating gamma and hadron-initiated EASs and measuring the nuclear mass composition of PCRs. There is a secondary data recording system to record the angular muon flux when no EAS trigger exist. The EASs predominantly produce these muons in the energy range of 10 GeV–10 TeV. The G3MT collects about 4 billion muons per day. This particular measurement is an ideal choice for studies of transient events such as thunderstorms and solar storms, cosmic ray modulation in the interplanetary space, etc. More details about the detector instrumentation can be found here [2].

## 3   Physics results

The primary objectives of GRAPES-3 experiment span over many orders of magnitudes in energy, starting from 1 GeV to 10 PeV. These objectives can be classified into their respective physics domains, namely (i) atmospheric acceleration, (ii) solar studies, and (iii) cosmic ray studies. Some published and preliminary results in the above categories are discussed briefly in the following subsections.

### 3.1   Atmospheric acceleration

Thunderstorm studies are emerging as one of the exciting areas of physics using cosmic ray secondaries. Especially muons are the ideal choice for studying these phenomena since they lose only a small and constant energy loss by ionization. One of the biggest mysteries in this field is the development of more than a billion volts in the thundercloud, which was predicted by C.T.R. Wilson almost a century ago [3]. The G3MT records about 50 significant thunderstorm events every year. One of the biggest thunderstorm events was recorded on 1 December 2014 that lasted for 18 minutes. The muon intensity dropped in 45 contiguous directions out of 169 directions. By combining the muon flux from those 45 directions, a clear deficit of 2% was seen with a significance of $10\sigma$ whereas the total significance was about $20\sigma$. Detailed Monte Carlo simulations allowed us to estimate the peak potential of the thundercloud to be (0.90±0.08) GV. Subsequent analyses on the shorter time scale of 2-minute muon exposure gave a conservative estimate of peak potential of 1.3 GV [4]. Clear evidence of cloud movement from east to west was seen in the 2-minute exposure map and in electric field measurements that estimate linear and angular velocities to be 1 km·min$^{-1}$ and 6.2° min$^{-1}$, respectively. The cloud height was estimated to be 11.4 km by combining linear and angular velocities. Considering the cloud coverage was in the entire field of view (FOV) of G3MT, the thundercloud should have a radius $\geq$11 km, implying an area of $\geq$380 km$^2$. Similarly, we estimated the electrical properties of the cloud by assuming a parallel-plate capacitor with an effective capacitance of $\geq$0.85 $\mu$F. A peak potential of 1.3 GV would require a total charge of $\geq$1100 C and energy of $\geq$720 GJ. The muon intensity variation has a rise time of 6 min, which implies a power delivery of $\geq$2 GW.

   Though the G3MT has been operated for more than two decades, the thunderstorm studies may be carried out using electric field measurements only since April 2011 after installing four electric field mills. A total of 487 significant thunderstorm events were identified from April 2011 to December 2020. These events were selected when $\Delta I_\mu \geq$0.3% and synchronous time variations in the electric field measurements were observed. These events show a clear asymmetry in their direction when distributed over a coarser 9-direction configuration, as shown in Table 1. About 80% of the total events were recorded in the east. There is a clear asymmetry in the east-west compared to the north-south orientation. This effect can be very well understood by the muon charge ratio present in nature. Figure 2 shows the muon charge ratio derived from the Monte Carlo simulations, where it can be clearly seen that the ratio is

| 6.2% | 1.8% | 30.1% |
|------|------|-------|
| (NW) | (N)  | (NE)  |
| 0.6% | 0.2% | 2.8%  |
| (W)  | (V)  | (E)   |
| 7.0% | 2.8% | 48.6% |
| (SW) | (S)  | (SE)  |

Table 1: The table consists of the percentage of thunderstorm events in 9-direction configuration of the G3MT FOV. The table is populated using 487 significant thunderstorm events collected from April 2011 to December 2020. Events in the vertical (V) direction are at the center of the table.

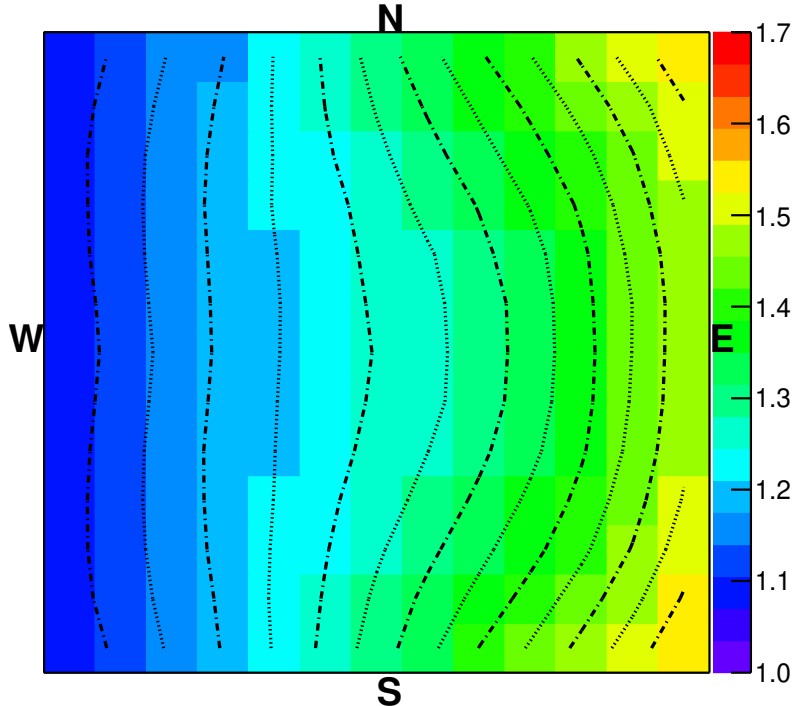

Figure 2: Muon charge ratio derived from Monte Carlo simulations in 169-direction configuration of the G3MT FOV.

higher in the east compared to the west. This particular asymmetry in the muon charge ratio results from bending of muons in the presence of geomagnetic field.

## 3.2   Solar studies

As explained in the previous section, measurement of angular muon fluxes $\geq$GeV is an ideal choice for studying transient events. On 22 June 2015, a series of coronal mass ejections (CMEs) were released from the surface of the Sun. Especially the third CME had a jump of >300 km·s$^{-1}$ in the solar wind velocity ($V_{SW}$) and triggered a G4 class geomagnetic storm. During that time the $B_z$ component of the interplanetary magnetic field (IMF) had a specific structure which resulted in a short muon burst recorded in the G3MT for 2 hours (Ref. Figure 1 & 2 of [5]). This muon burst is believed to be caused by the magnetic reconnection of IMF $B_z$ with the geomagnetic field (GMF), which lowered the cutoff rigidity for incoming PCRs. The entry of excess low-energy PCRs produced more EASs resulting in a muon burst. Also, the muon burst was observed simultaneously in all nine directions, which indicates that this

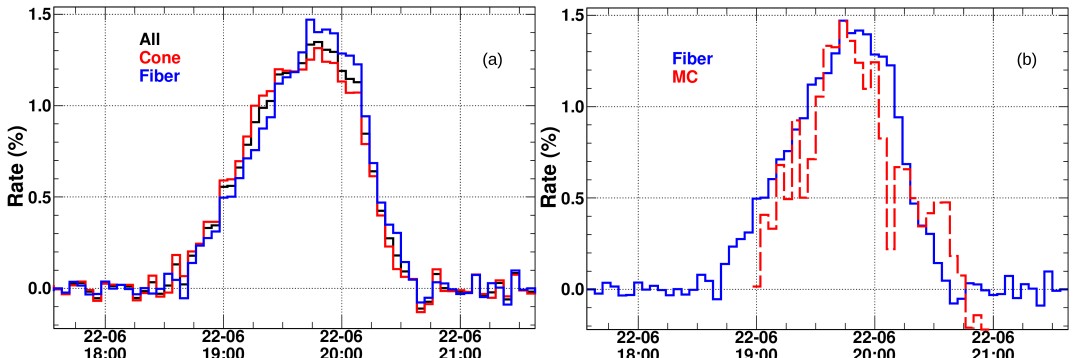

Figure 3: The figures show (a) background corrected scintillator rate showing detection of excess count rates by the selected detectors for 22 June 2015 event. The profiles are also shown for different types of detectors namely cone and fiber; (b) background corrected scintillator rate for fiber detector in comparison with Monte Carlo simulation.

effect was localized (Ref. Figure 3 of [5]). Detailed Monte Carlo studies confirmed that the observed phenomena were indeed due to the lowering of cutoff rigidity by the interaction of IMF with GMF [5]. Interestingly the muon burst was observed 32 minutes after the arrival of IMF. Studying such geomagnetic storm events may help us to understand its propagation and effects in the interplanetary medium. As mentioned before, the G3MT has been continuously operated over two decades. We recorded about 80 such geomagnetic storm events having various amplitudes and delays during this period. A multi-parameterization study involving the observed muon intensity and the solar wind parameters measured at the L1 point may help better understand future solar storms.

In a recent study, we found that geomagnetic storm events were also recorded in the scintillator detectors. Each scintillator detector counts the number of particles ($\sim$200–300 sec$^{-1}$) above a certain threshold (few MeVs). Here, the detection includes particles such as muon, gamma, electron, hadron, etc. Unlike G3MT, the scintillators do not record the direction of the passing particle. We noted that scintillator rates are prone to temperature effects due to the photomultiplier tubes used. However, one can make a quantitative selection of detectors having less temperature dependence for a better signal-to-noise ratio. Figure 3a shows the background corrected scintillator rates for selected detectors after applying stringent cuts. Figure 3b shows the background corrected scintillator rate for the fiber detector compared to Monte Carlo simulation. It is quite interesting to note that the scintillator rate has recorded $\sim$40% higher amplitude than G3MT. The Monte Carlo simulations found that the recorded scintillator rate count was composed of 58% muons, 11% gamma, 29% electromagnetic components, and the remainder hadrons. The estimated scaling factor and delay are consistent with the G3MT's observation. This particular data may allow identifying weak geomagnetic storm events.

## 3.3   Cosmic ray studies

### 3.3.1   Improvements in angular resolution

A precise reconstruction of EAS direction is an essential aspect in studies of CR origin and gamma-ray astronomy. It is well understood that the shower front has a curvature. Conventionally it has been corrected by applying a constant curvature (0.215 ns·m$^{-1}$) to the shower front, and then a planar fit is performed to estimate the EAS direction. In a recent study, it was found that the shower front curvature has a strong dependence on shower size and shower

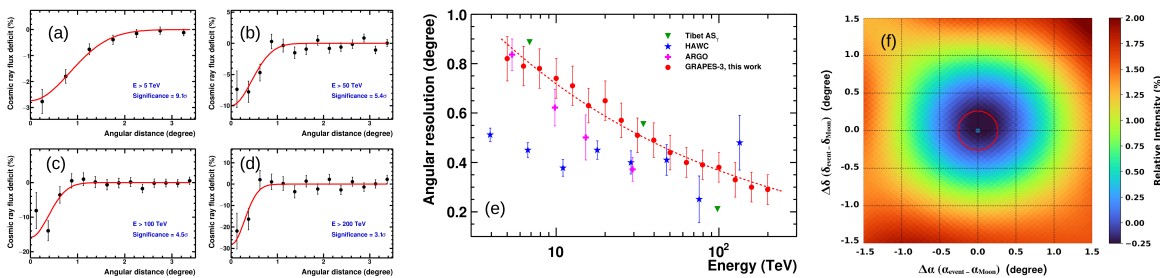

Figure 4: The figures show (a)–(d) the cosmic ray shadow of the Moon for energies above 5, 50, 100, and 200 TeV respectively; (e) the angular resolution of the GRAPES-3 array compared with other EAS experiments; (f) the pointing accuracy of the GRAPES-3 array along the right ascension ($\alpha$) and declination ($\delta$) are estimated to be $(0.032\pm0.004)°$ and $(0.090\pm0.003)°$ respectively.

age [6]. In this work, the curvature dependences were studied in great detail and corrected to get an improved fit. We achieved an angular resolution starting from $0.78°$ at $>5$ TeV to $0.17°$ at $>500$ TeV. These improvements were also achieved because of better time measurement using HPTDC and real-time estimation of time offset using a statistical method called "random walk". The GRAPES-3's angular resolution is compared with other experiments such as ARGO-YBJ, Tibet AS$\gamma$, and HAWC (Figure 15 of [6]). ARGO-YBJ reported an overall angular resolution of better than $0.5°$ in the energy range of 5–30 TeV [7]. Tibet AS$\gamma$ reported an angular resolution of $0.87°$ at 7 TeV, improving to $0.54°$ at 35 TeV [8,9]. Similarly, HAWC reported that the angular resolution is better than $0.5°$ in the energy range of 5–100 TeV [10]. These measurements are comparable to GRAPES-3. However, these experiments are located at an altitude above 4000 m which is approximately twice the altitude of GRAPES-3. It is expected to have a factor improvement in the angular resolutions of other experiments located at higher elevations, considering their shallower depths [11]. The other experiments may also gain an improvement in the angular resolution if the size and age dependent curvature correction is applied.

### 3.3.2   Cosmic ray shadow of the Moon

Another important aspect of EAS arrays is understanding the direction reconstruction's pointing accuracy. One reliable and widely accepted method is to use the cosmic ray shadow of the Moon. The Moon is a big obstacle with an angular diameter of about $0.5°$ for incoming PCRs. A study of this shadowing effect using the EAS array helps to calibrate its angular resolution and pointing accuracy. We used the EAS data collected during 2014–2016. Figures 4a–d show the Moon shadow as a function of angular distance from the center of the Moon for energies above 5, 50, 100, and 200 TeV. The angular resolution using this method was estimated to be $(0.83\pm0.09)°$ at $>5$ TeV with a significance of $9.1\sigma$ (Figure 4a). This improves to $(0.29\pm0.06)°$ at $>200$ TeV with a significance of $3.1\sigma$ (Figure 4d). These results are consistent with the earlier analysis [6] described in the previous subsection and comparable with other experiments that are located at almost twice the altitude of GRAPES-3 (Figure 4e). We estimated the pointing accuracy of the EAS array along right ascension ($\alpha$) and declination ($\delta$) to be $(0.032\pm0.004)°$ and $(0.090\pm0.003)°$, respectively (Figure 4f). More details can be found in the detailed report [12].

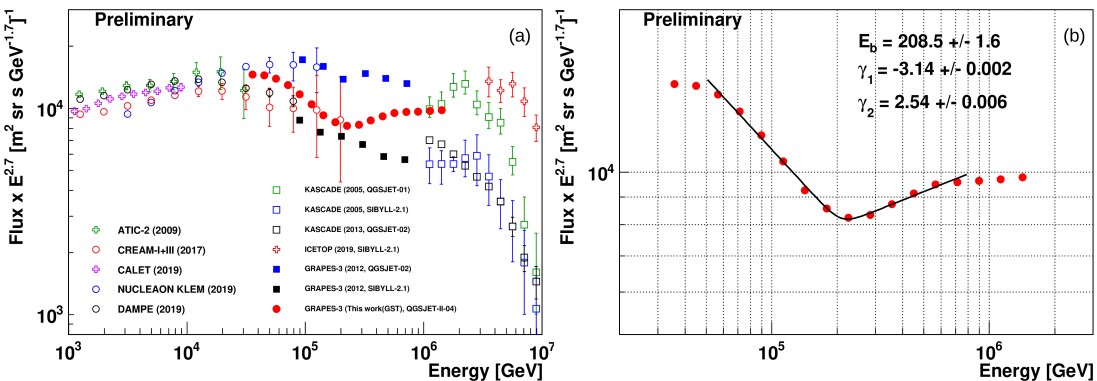

Figure 5: The figures show (a) proton spectrum compared with measurements from other experiments and (b) a spectral break at $\sim$208 TeV.

### 3.3.3   Primary energy spectrum and mass composition

As mentioned earlier, the GRAPES-3's energy spectrum measurement in the energy range of 1 TeV–10 PeV may provide an overlap between low and ultra-high energy measurements from other experiments. For the preliminary study, $1.47\times10^7$ EASs recorded from January 2014 to October 2015 were selected after imposing quality cuts to enrich the data quality. The EASs were selected with the following quality cuts: (i) successful direction and Nishimura-Kamara-Greisen reconstruction, (ii) Cores within the fiducial area (7850 $m^2$) of 50 m radius from the center of the array, (iii) shower age 0.2<s≤1.8, (iv) zenith $\theta$<18°, and (v) shower size $N_e>10^4$, at which the trigger efficiency is above 90%. The selected EASs were translated from shower size to energy with the aid of Monte Carlo simulations. CORSIKA v76900 was used with QGSJETII-04 and FLUKA for high and low-energy hadronic interaction models. The EASs were simulated in the energy range of 1 TeV–10 PeV with spectral index $\gamma$=–2.5 for mass species proton, helium, nitrogen, aluminium, and iron. Each mass type has $1.2\times10^8$ simulated EASs. We restricted the zenith angle to 45°. The Monte Carlo data set was subjected to detector simulation and reconstructed to get the primary properties for the experimental data. Figure 5a shows the measured proton spectrum compared with other measurements from direct [13–17] and indirect experiments [18–20]. We can see that the GRAPES-3 proton spectrum has a reasonably good overlap with other measurements. A notable feature can be seen at 208.5±1.6 TeV where the spectrum hardens from $\gamma_1$=–3.14 to $\gamma_2$=–2.54. More details may be found in the contributed talk in this symposium (F. Varsi et al. "Updated results on the cosmic ray energy spectrum and composition from the GRAPES-3 experiment").

## 4   Conclusion

Some published and preliminary results of GRAPES-3 were presented during the symposium covering atmospheric acceleration, solar studies, energy spectrum and composition, angular resolution, cosmic ray anisotropy, etc. The muon imaging technique allowed us to measure 1.3 GV electric potential in one of the massive thunderclouds recorded by the muon telescope, providing many insights into the electrical and geometrical properties of thunderclouds. A collection of 487 significant thunderstorms indicated a clear directional asymmetry which the muon charge asymmetry can explain. Similarly, the geomagnetic studies and their implications on the cosmic ray flux and identification of many such events using the GRAPES-3 muon telescope may provide key inputs in the advancement of space weather prediction. Recent studies revealed that the GRAPES-3 scintillators also provide vital information in understanding geo-

magnetic storms, especially those with weak signals that the muon telescope can not detect. Earlier studies on the angular resolution of the GRAPES-3 EAS array were validated using the cosmic ray shadow of the Moon. The angular resolution was estimated to be (0.83±0.09)° at >5 TeV and improves to (0.29±0.06)° at >200 TeV, confirming the earlier studies using different techniques. This angular resolution is comparable to other experiments that are located at almost twice the altitude of GRAPES-3 [7–10]. The pointing accuracy was estimated to be (0.032±0.004)° and (0.090±0.003)° along the right ascension and declination, respectively. Considering the EAS data collected from January 2014 to October 2015, the proton energy spectrum was obtained employing Monte Carlo simulations using CORSIKA. The measured spectrum was found to have a reasonably good overlap with other measurements. We found a notable spectral break at ∼208 TeV. Currently, the GRAPES-3 muon telescope is being upgraded to double its area and sensitivity, which is expected to improve its physics potential.

## Acknowledgements

We thank D.B. Arjunan, A.S. Bosco, V. Jeyakumar, S.Kingston, K. Manjunath, S. Murugapandian, S. Pandurangan, B. Rajesh, V. Santhoshkumar, M.S. Shareef, C. Shobana, and R. Sureshkumar for their efforts in maintaining the GRAPES-3 experiment.

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
