# Peer review of "Highlights of the results from the GRAPES-3 experiment"

_SciPost Physics Proceedings_

## Round 1 · Referee Report · Eduardo de la Fuente Acosta (Referee 1) · 2022-9-30

Strengths

  1. Science
  2. Experiment

Weaknesses

  1. Comparison and Discussion 2 English

Report

The quality and description of the experiment are outstanding, and the work deserves publication. However, the authors must improve the English of the manuscript to get a more quality paper. I strongly suggest a native English speaker or Angloparlant must review the manuscript. I gave some suggestions (in red) in the attached pdf file. This file also includes my suggestions and comments in blue.

The only significant comment is on the angular resolution issue. The authors compare it with other successful experiments. In some sentences, they said it is better, and in another sentence, they commented it is comparable. The authors must choose the right one. If the angular resolution is better, the issue must be clarified and described in more detail. In case of comparison, values of other experiments are needed to add for contrast. However, for all comparisons, the angular resolution value of all experiments must be mentioned and concluded after contrasting them.

Congratulations to the authors for the experiment, strengthen them to provide a better version of the paper and improved.

Requested changes

  1. English
  2. Work on comments about angular resolution

Attachment

---

## Round 2 · Author Response

List of changes
1) Comment: "The quality and description of the experiment are outstanding, and the work deserves publication. However, the authors must improve the English of the manuscript to get a more quality paper. I strongly suggest a native English speaker or Angloparlant must review the manuscript. I gave some suggestions (in red) in the attached pdf file. This file also includes my suggestions and comments in blue."
Reply: We thank the referee for the critical comments and suggestions to improve the manuscript. All the suggestions and comments are implemented. It was also reviewed and corrected by a native English speaker.
2) Comment: "The only significant comment is on the angular resolution issue. The authors compare it with other successful experiments. In some sentences, they said it is better, and in another sentence, they commented it is comparable. The authors must choose the right one. If the angular resolution is better, the issue must be clarified and described in more detail. In case of comparison, values of other experiments are needed to add for contrast. However, for all comparisons, the angular resolution value of all experiments must be mentioned and concluded after contrasting them."
Reply: We thank the referee for bringing this critical suggestion. We rephrased the sentences and added other experiments' values. Furthermore, we added references to other experiments' work. We would also like to clarify that there are two different results from GRAPES-3 on angular resolution discussed, namely (i) Section 3.3.1 [V.B. Jhansi et al., Journal of Cosmology and Astroparticle
Physics 07, 024 (2020)] and (ii) Section 3.3.2 [D. Pattanaik et al., Phys. Rev. D 106, 022009 (2022)]. Thus, the quoted values have small differences, but within error.

---

## Round 2 · List of Changes

1) Comment: "The quality and description of the experiment are outstanding, and the work deserves publication. However, the authors must improve the English of the manuscript to get a more quality paper. I strongly suggest a native English speaker or Angloparlant must review the manuscript. I gave some suggestions (in red) in the attached pdf file. This file also includes my suggestions and comments in blue."
Reply: We thank the referee for the critical comments and suggestions to improve the manuscript. All the suggestions and comments are implemented. It was also reviewed and corrected by a native English speaker.
2) Comment: "The only significant comment is on the angular resolution issue. The authors compare it with other successful experiments. In some sentences, they said it is better, and in another sentence, they commented it is comparable. The authors must choose the right one. If the angular resolution is better, the issue must be clarified and described in more detail. In case of comparison, values of other experiments are needed to add for contrast. However, for all comparisons, the angular resolution value of all experiments must be mentioned and concluded after contrasting them."
Reply: We thank the referee for bringing this critical suggestion. We rephrased the sentences and added other experiments' values. Furthermore, we added references to other experiments' work. We would also like to clarify that there are two different results from GRAPES-3 on angular resolution discussed, namely (i) Section 3.3.1 [V.B. Jhansi et al., Journal of Cosmology and Astroparticle
Physics 07, 024 (2020)] and (ii) Section 3.3.2 [D. Pattanaik et al., Phys. Rev. D 106, 022009 (2022)]. Thus, the quoted values have small differences, but within error.

---

## Editorial Decision

editorial_decision: